# Comparison of the Learning Curve between Uniportal and Robotic Thoracoscopic Approaches in Pulmonary Segmentectomy during the Implementation Period Using Cumulative Sum Analysis

**DOI:** 10.3390/cancers16010184

**Published:** 2023-12-29

**Authors:** Hitoshi Igai, Kazuki Numajiri, Fumi Ohsawa, Kazuhito Nii, Mitsuhiro Kamiyoshihara

**Affiliations:** Department of General Thoracic Surgery, Japanese Red Cross Maebashi Hospital, 389-1 Asakura-cho, Maebashi 371-0811, Gunma, Japan

**Keywords:** uVATS, RATS, learning curve, CUSUM, segmentectomy

## Abstract

**Simple Summary:**

There have been few studies comparing the perioperative outcomes and learning curves of the operative time in pulmonary segmentectomy between uVATS and RATS performed by the same single surgeon, although both approaches are emerging as minimally invasive techniques for major lung resections worldwide. This study revealed that a surgeon reached the proficiency period earlier in RATS than in uVATS, although the trends to the stable period were similar. In contrast, most perioperative outcomes were statistically similar between the two approaches. When a surgeon begins each approach in pulmonary segmentectomy, the difference in operative time trend should be considered to achieve successful implementation.

**Abstract:**

Background: The aim of this retrospective study was to compare the learning curve and perioperative outcomes between the two approaches uVATS and RATS during their implementation periods. Methods: The uVATS group included 77 consecutive uVATS segmentectomies performed by HI between February 2019 and June 2022, while the RATS group included 30 between July 2022 and September 2023. The patient characteristics, perioperative outcomes, and learning curves were compared between the two groups. The learning curve was evaluated using operative time and cumulative sum (CUSUM_OT_) analysis. Results: Most patient characteristics and perioperative outcomes were equivalent between the two groups. In the uVATS group, after a positive slope was observed until the 14th case (initial period), a plateau was observed until the 38th case (stable period). Finally, a negative slope was observed after the 38th case (proficiency period). In the RATS group, after a positive slope was observed until the 16th case (initial period), a plateau was observed until the 22nd case (stable period). Finally, a negative slope was observed after the 22nd case (proficiency period). Conclusions: In segmentectomy, a surgeon reached the proficiency period earlier in RATS than in uVATS, although the trends to the stable period were similar.

## 1. Introduction

Uniportal and robotic thoracoscopic approaches are emerging as minimally invasive techniques for major lung resections worldwide, while the conventional multiportal video-assisted thoracic surgery (VATS) approach has reached maturity. After Rocco et al. first reported wedge resection using the uniportal VATS (uVATS) approach in 2004, Gonzalez et al. first introduced this approach for major lung resection in 2011 [1,2]. This less invasive approach through a single skin incision has gained worldwide acceptance. Several reports have suggested that the uniportal thoracoscopic approach for major lung resection, in addition to a smaller skin incision, may offer additional superior perioperative outcomes, including a significantly shorter operative time, less intraoperative bleeding, and shorter postoperative drainage or hospital stay, compared with the multiportal approach [3,4,5].

On the other hand, the robotic-assisted thoracoscopic surgery (RATS) approach for lobectomy was first reported by Melfi et al. in 2002 [6]. High-definition 3D views and an ergonomic design are considered as advantages of RATS compared to conventional VATS. Since its introduction, several previous articles have described the successful perioperative outcomes of RATS anatomical lung resections [7,8,9]. 

With the increasing trend toward minimally invasive surgery, segmentectomy has been used as an alternative to lobectomy in early-stage NSCLC. In addition, two randomized controlled trials (JCOG0802/WJOG4607L and CALGB140503) confirmed the non-inferiority of sublobar resections, including segmentectomy, to lobectomy in patients with stage IA NSCLC (tumor diameter ≤ 2 cm) [10,11], which means that segmentectomy is likely to be increasingly performed in the future. Therefore, segmentectomy should be appropriately performed by both uVATS and RATS.

At our institution, uVATS for anatomical segmentectomy was introduced in February 2019, and RATS was introduced in June 2022. During the implementation phase of new surgical approaches, wider adoption can be facilitated by identifying how many cases are needed to achieve proficiency. The aim of this retrospective study was to compare the learning curve and perioperative outcomes between the two approaches during the implementation period.

## 2. Patients and Methods

The study was conducted in accordance with the Declaration of Helsinki (as revised in 2013). The study protocol was approved by the institutional ethics board of Japanese Red Cross Maebashi Hospital (Approval No.: 2023-45) and the need for individual consent for this retrospective analysis was waived.

Between February 2019 and September 2023, 213 patients underwent pulmonary segmentectomy via a minimally invasive approach at our institution. Between February 2019 and June 2022, HI performed uVATS segmentectomy in 77 consecutive cases, which were included in the uVATS group. After this period, all the segmentectomies in our department were performed by other surgeons because RATS segmentectomy was introduced, which was performed by HI. Therefore, the RATS group included 30 consecutive cases between July 2022 and September 2023. Figure 1 shows the patient enrollment process. HI had performed more than 500 conventional multiportal VATS major lung resections before starting to use uVATS. Subsequently, HI started RATS for major pulmonary resections after performing nearly 150 uVATS major pulmonary resections with no RATS thymectomy experience.

The clinical data analyzed for each case included age, sex, lobe treated, American Society of Anesthesiologists (ASA) score, smoking index (pack-years), forced expiratory volume in one second (FEV1.0), %FEV1.0, disease, type of segmentectomy, operative time, blood loss, postoperative drainage time, postoperative hospitalization time, morbidity (Clavien–Dindo grade ≥ III), readmission within 30 days after surgery, conversion to thoracotomy, 30-day postoperative mortality. All segmentectomies were classified into simple and complex types [12]. Simple pulmonary segmentectomy included the lingual, basilar, or superior segment of the lower lobe or the upper division of the left upper lobe. Complex pulmonary segmentectomy was defined as any pulmonary segmentectomy other than those mentioned above.

### 2.1. Surgical Procedures 

Both approaches were performed with the patient in the lateral decubitus position under general anesthesia and with single-lung ventilation. 

In the uVATS approach, a single 3.5–4 cm skin incision was made at the anterior axillary line of the 4th or 5th intercostal space. The incision was initially covered with an extra-small wound retractor (Alexis Wound Retractor; Applied Medical, Rancho Santa Margarita, CA, USA). All surgical instruments and a 10 mm, 30-degree-angled thoracoscope were simultaneously inserted through the incision. Figure 2A shows a single skin incision and operative findings using the uVATS approach.

In the RATS (the DaVinci XI robotic system; Intuitive, Sunnyvale, CA, USA) approach, five ports, including an assistant port, were used. Figure 2B shows the position of each port in the RATS approach. The cavity was evaluated with a 0-degree-angled camera. Three 8 mm robotic trocars were placed in the 6th or 7th intercostal space, centered on the scapular line. For retraction or stapler insertion, a 12 mm robotic trocar was placed at the anterior axillary line of the 7th or 8th intercostal space. An assistant port was placed at the scapular line of the 9th intercostal space. A CO_2_ insufflation pressure of 8 mmHg was used.

The surgical manners were similar for both approaches. Dominant vessels, including the pulmonary artery and vein, were adequately exposed and then divided, mainly using endovascular staplers. In the RATS approach, robotic endostaplers were always chosen. Small branches of these vessels were divided with an energy device after proximal ligation with silk sutures. The dominant bronchus was also divided with a stapler. Intersegmental planes were identified using infrared thoracoscopy with intravenous administration of indocyanine green. Alternatively, inflation–deflation was used in patients with an iodine allergy. All intersegmental planes were divided with staplers. If an air leak was detected during a seal test at the end of surgery, the leak was sutured with an absorbable monofilament or sealant was applied. In patients with primary lung cancer who underwent segmentectomy, interlobar and hilar lymph node sampling was performed for pathologic stage confirmation. If the resected lymph node was positive in intentional segmentectomies, we planned to perform an additional lobectomy. No intraoperative frozen section of the lymph nodes was examined because cT1aN0 was the criteria for intentional segmentectomies in our department. For unintentional segmentectomies, we did not plan to perform an additional lobectomy. These patients could not tolerate an additional lobectomy. Patients with metastatic or benign disease did not undergo lymphadenectomy. A chest tube was put in the thoracic cavity.

### 2.2. Postoperative Treatment

The chest tube was removed after confirmation that there was no active bleeding and no air leak. From February 2019 to June 2021, the tube was left at least until postoperative day 1. However, from July 2021, we started early chest tube removal on the day of surgery [13]. Indications for drain removal on the day of surgery were an absence of air leakage during an intraoperative seal test, radiographic evidence of lung expansion, and continuous absence of air leakage through a drainage bottle for 2–4 h after surgery. Patients could be discharged if the chest X-ray taken the day after chest tube removal showed no problems. Postoperative complications were evaluated according to the Clavien–Dindo classification [14].

### 2.3. Evaluation of the Learning Curve

The cumulative sum (CUSUM) method, which is the cumulative sum of the differences between each data point and the mean of all data points, was used to quantitatively assess the learning curve. The CUSUM method allows for the detection of small changes in performance measures that may not be detectable using other measures [15,16]. The CUSUM for the variables of interest in the first patient was the difference between the value for the first patient and the mean for all patients. The CUSUM for the second patient was the CUSUM for the previous patient added to the difference obtained for the second patient. This recursive process continued until the CUSUM for the last patient was calculated as zero.

In this study, the learning curve was first evaluated using the operative time and CUSUM (CUSUM_OT_). We evaluated the curve of best fit to detect the change in slope of the CUSUM_OT_ learning curve. In this method, positive and negative slopes indicated a series of cases with above-average and below-average operative times, respectively. The number of cases required for learning was calculated from the inflection point of the curve of the best-fit line.

### 2.4. Statistical Analysis

The Mann–Whitney U test for continuous variables or Fisher’s exact test for categorical variables was used to assess patient characteristics and perioperative outcomes between the two approaches. Differences were considered significant at *p* < 0.05. All calculations and statistical analyses were performed using the EZR graphical user interface for R (version 1.40, Saitama Medical Centre, Jichi Medical University, Saitama, Japan).

## 3. Results 

### 3.1. Patient Characteristics and Perioperative Outcomes

Table 1 shows the comparison of patient characteristics and perioperative outcomes between the two approaches. In the characteristics, although the proportion of disease was significantly different, all other variables were similar between the two groups. In the perioperative outcomes, postoperative drainage time (uVATS group: 1 [IQR: 1–1] vs. RATS group: 0 [IQR: 0–1] day, *p* < 0.0001) and hospital stay (uVATS group: 2 [IQR: 2–3] vs. RATS group: 2 [IQR: 1–3] days, *p* = 0.0018) were significantly shorter in the RATS group than in the uVATS group, although other perioperative outcomes including operative time were equivalent between the two groups. However, a subset analysis including only the patients from July 2021, when we started early chest tube removal on the day of surgery, to September 2023 showed that postoperative drainage time (uVATS group: 1 [IQR: 0–1] vs. RATS group: 0 [IQR: 0–1] day, *p* = 0.27) and hospital stay (uVATS group: 2 [IQR: 2–2] vs. RATS group: 2 [IQR: 1–3] days, *p* = 0.45) in the RATS group were not significantly different (Table 2). Only two cases in the uVATS group had postoperative morbidity, consisting of paroxysmal atrial fibrillation and a prolonged air leak.

### 3.2. Distributions of the Performed Segmentectomies in Both Groups

Table 3 shows the details of the segmentectomies performed between the two groups. The proportions of the types of segmentectomies, including simple and complex, were statistically similar between the two groups (*p* = 0.28).

### 3.3. Learning Curves

Figure 3 shows the raw operating time data and the CUSUM_OT_ learning curve in the uVATS group. The CUSUM_OT_ learning curve was best modeled as a third-order polynomial with the CUSUM equation in minutes equal to 0.0035 × case number^3^ − 0.5084 × case number^2^ + 20.092 × case number^1^ − 104.96. After a positive slope was observed until the 14th case (initial period), a plateau slope was observed until the 38th case (stable period), meaning that the operative time became stable. Finally, a negative slope, meaning that the operative time became shorter than the average, was observed after the 38th case (proficiency period). 

Figure 4 shows the raw operating time data and the CUSUM_OT_ learning curves in the RATS approach. The CUSUM_OT_ learning curve was best modeled as a third-order polynomial with the CUSUM equation in minutes equal to −0.0498 × case number^3^ + 1.6764 × case number^2^ − 7.212 × case number^1^ + 6.9162. After a positive slope was observed until the 16th case (initial period), a plateau slope was observed until the 22nd case (stable period), which meant that the operating time became stable. Finally, after the 22nd case, a negative slope was observed, meaning that the operative time became shorter than the average (proficiency period).

## 4. Discussion

In this study, our group described the comparison of perioperative outcomes between uVATS and RATS segmentectomies performed by a single surgeon. In addition, the learning curve of the operative time in each approach was evaluated using CUSUM analysis. Few reports have focused on the comparison of perioperative outcomes between uVATS and RATS anatomical resections [17,18]. Yang et al. described early results of robotic versus uniportal video-assisted thoracic surgery for lung cancer and concluded that robotic surgery resulted in less intraoperative blood loss and more dissected lymph nodes, although other perioperative outcomes were equivalent [17]. Zhang et al. also showed that the number of harvested lymph nodes was higher with robotic surgery, while postoperative drainage and hospital stay were shorter with the uniportal approach [18]. Although the results were valuable, most of the patients included in these studies underwent lobectomy. In addition, the learning curve was not evaluated. To the best of our knowledge, this is the first study to compare the perioperative outcomes and learning curves of operative time in pulmonary segmentectomy between uVATS and RATS performed by the same single surgeon.

There have been several studies evaluating the learning curves of pulmonary segmentectomies in each approach [19,20,21,22,23,24,25,26,27,28]. Lei et al. reported that the learning curve of the operative time in uniportal thoracoscopic segmentectomy was divided into three phases among 120 patients, and the surgeon reached the mastery phase in 60 cases [19]. Zhang et al. reported that the learning curve of the operative time became the mastery phase after the surgeon experienced 46 cases in robotic pulmonary segmentectomy among 104 cases [24]. Le Gac et al. demonstrated that the learning curves of operative time in robotic segmentectomy were divided into two phases, and the inflection point was the 27th case [23]. They also indicated that the cost was reduced with an improved learning curve. Although the learning curve of operative time was evaluated using CUSUM analysis in most of the previous studies, it was often divided into three distinct phases. The negative slope in the third period was considered as the first proficiency. However, Li insisted that the fourth proficiency stage in the learning curve of uniportal thoracoscopic lobectomy was entered after the experience of 244 procedures [29]. Therefore, our learning curve of pulmonary segmentectomy through each approach might change in the future after gaining experience.

Although the number of cases required to reach the stable period was almost similar between the two groups, more cases were required to reach the proficiency period in the uVATS group than in the RATS group. This result suggests that although it takes a similar number of cases to reach the stable period in each approach, surgeons become proficient in RATS faster than in the uVATS approach. In our RATS approach, five ports were used, which meant that the surgical principle was the same as the conventional multiportal thoracoscopic approach. On the contrary, the uVATS approach required specific techniques including non-grasping or suction-guided stapling to perform the operation, which were quite different from the conventional multiportal thoracoscopic approach [30,31]. Therefore, a difference in the number of cases required between the two groups might occur. Furthermore, Andersson et al. reported that robotic lobectomy can be performed safely and efficiently in an expert center with previous experience in conventional multiportal thoracoscopic lobectomy, suggesting that high-definition 3D view, minimal hand tremor, and improved surgeon ergonomics may facilitate the learning curve [32].

We evaluated the learning curves for pulmonary segmentectomies using the uVATS and RATS approaches by operative time. However, learning curves can also be evaluated by the rate of surgical complications. Gómez-Hernández et al. used an unadjusted CUSUM to evaluate surgical complications in anatomical lung resections by VATS and RATS and, finally, showed that the inflection point was the 28th case in uVATS and the 32nd case in RATS, which was concluded to be a similar number of cases [33]. We fully agree with this evaluation because the rate of surgical complications reflects the operative quality and should be reduced to maintain the minimally invasive nature. In the present study, we did not evaluate this because surgical complications, all of which were significant vascular injuries, occurred only in three cases in the uVATS group and did not occur in the RATS group. In the future, we will try to evaluate it after obtaining the cases.

In pulmonary segmentectomy, postoperative prolonged air leak (PAL) occasionally occurred, especially in complicated type of segmentectomy [34]. In most cases, the origin of an air leak is considered to be the staple line of the divided intersegmental plane, which indicates that proper stapling is quite important to avoid postoperative PAL in pulmonary segmentectomy. Our results showed that there were no cases of postoperative PAL in both groups, and only one patient in the uVATS group suffered from delayed pulmonary fistula, which indicated that the appropriate stapling was performed in both approaches, although the direction of the inserted stapler was limited. 

## 5. Limitations

The current study had several limitations. First, it involved a single institution and was retrospective in nature, with a small number of enrolled patients. Second, the period when each approach was performed was different and the RATS approach started later than uVATS, which meant that HI had more surgical experience when he started the RATS approach compared to the uVATS approach. This might affect the positive result that HI reached the proficiency period earlier in the RATS approach than in the uVATS approach. Third, the distribution of segmentectomies performed differed between the two groups. While most types of pulmonary segmentectomies were performed in the uVATS approach, several types of them, including lingual (S4–5) or lateral and posterior basal (S9–10), were not performed in the RATS approach. Fourth, the emergence of new equipment and the presence of different surgical nurses or assistants may affect perioperative outcomes. Finally, long-term outcomes including local recurrence rate were not evaluated.

## 6. Conclusions

Comparing the operative time for pulmonary segmentectomy between the uVATS and RATS approaches, one surgeon reached the proficiency period earlier with RATS than with uVATS, although the trends toward the stable period were similar. In contrast, most perioperative outcomes were statistically similar between the two approaches. When a surgeon begins each approach to pulmonary segmentectomy, the trend in the difference in operative time should be considered to achieve successful implementation.

## Figures and Tables

**Figure 1 cancers-16-00184-f001:**
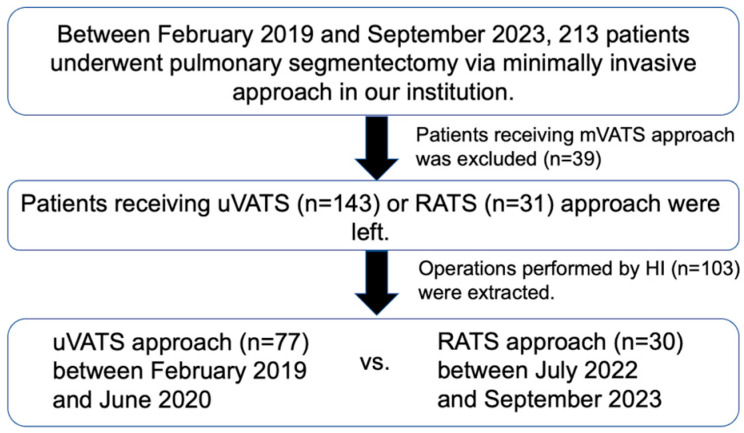
Outline of the patient enrollment process. mVATS: multiportal video-assisted thoracic surgery; uVATS: uniportal video-assisted thoracic surgery; RATS: robotic-assisted thoracoscopic surgery.

**Figure 2 cancers-16-00184-f002:**
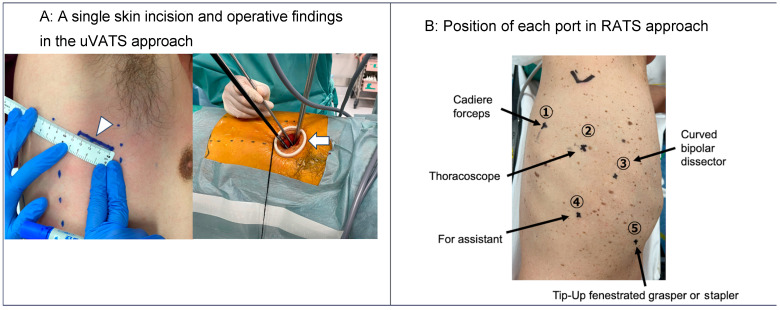
(**A**) A single skin incision and operative findings in the uVATS approach. A 4 cm single skin incision (arrowhead) was made at the 4th or 5th intercostal space of the anterior axillary line. All surgical instruments and a thoracoscope were simultaneously inserted through the incision (arrow). (**B**) Position of each port in the RATS approach. Five ports, including an assistant port, were used. uVATS: uniportal video-assisted thoracic surgery; RATS: robotic-assisted thoracoscopic surgery.

**Figure 3 cancers-16-00184-f003:**
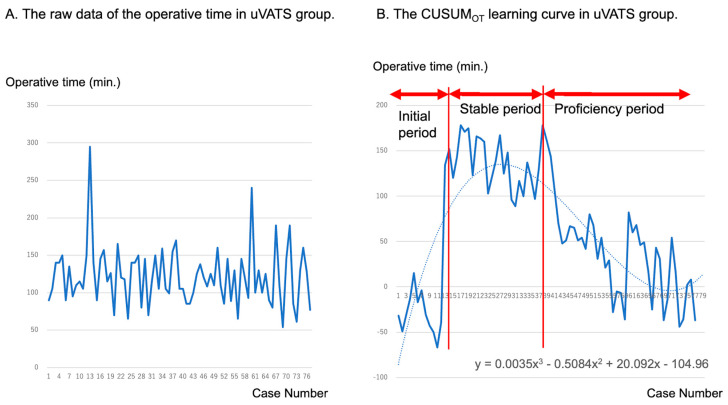
Raw data of operative times (**A**) and learning curves of operative time and cumulative sum (CUSUM_OT_) (**B**) in the 77 cases that underwent uVATS segmentectomy. CUSUM: cumulative sum method; uVATS: uniportal video-assisted thoracic surgery.

**Figure 4 cancers-16-00184-f004:**
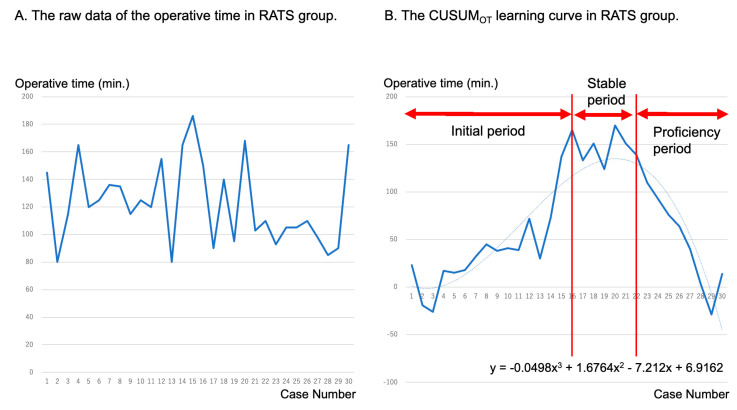
Raw data of operative times (**A**) and learning curves of operative time and cumulative sum (CUSUM_OT_) (**B**) in the 30 cases who received RATS segmentectomy. CUSUM: cumulative sum method; RATS: robotic-assisted thoracoscopic surgery.

**Table 1 cancers-16-00184-t001:** Comparison of the patient characteristics and perioperative outcomes between uVATS and RATS groups.

Variable	uVATS Group (*n* = 77)	RATS Group (*n* = 30)	*p*-Value
Age, years, median (IQR)	70 (66–77)	77 (62–79)	0.59
Sex			1
Female/male, *n* (%)	38 (49.4)/39 (50.6)	15 (50)/15 (50)	
Treated lobe			0.18
LUL, *n* (%)	21 (27.3)	15 (50)	
LLL, *n* (%)	12 (15.6)	4 (13.3)	
RUL, *n* (%)	24 (31.2)	6 (20)	
RML, *n* (%)	0 (0)	0 (0)	
RLL, *n* (%)	20 (26)	5 (16.7)	
ASA score, median (IQR)	2 (2–2)	2 (2–3)	0.23
Smoking index, pack-years,	15	3	0.56
median (IQR)	(0–46)	(0–39)	
Preoperative FEV1.0, mL,	1890	1915	0.89
median (IQR)	(1628–2390)	(1353–2808)	
Preoperative %FEV1.0, %,	92	87	0.65
median (IQR)	(77–103)	(69–107)	
Disease			0.0084
Primary lung cancer, *n* (%)	52 (67.5)	19 (63.3)	
Pulmonary metastasis, *n* (%)	9 (11.7)	9 (30)	
Other malignancy, *n* (%)	0 (0)	1 (3.3)	
Other benign, *n* (%)	16 (20.8)	1 (3.3)	
Type of segmentectomy			0.96
Intentional, *n* (%)	36 (46.8)	14 (46.7)	
Unintentional, *n* (%)	18 (23.4)	6 (20)	
Others, *n* (%)	23 (29.9)	10 (33.3)	
Operative time, minutes, median (IQR)	118 (95–145)	118 (99–144)	0.72
Blood loss, gram, median (IQR)	0 (0–30)	0 (0–0)	0.22
Postoperative drainage time, days, median (IQR)	1 (1–1)	0 (0–1)	<0.001
Postoperative hospitalization time, days, median (IQR)	2 (2–3)	2 (1–3)	0.0018
Morbidity (Clavien–Dindo classification grade ≥ 3), *n* (%)	2 (2.6)	0 (0)	1
Readmission within 30 days after surgery, *n* (%)	0 (0)	0 (0)	—
Conversion to thoracotomy, *n* (%)	2 (2.6)	0 (0)	1
30-day mortality, *n* (%)	0 (0)	0 (0)	—
90-day mortality, *n* (%)	0 (0)	0 (0)	—

IQR, interquartile range; LUL, left upper lobe; LLL, left lower lobe; RUL, right upper lobe; RML, right middle lobe; RLL, right lower lobe; ASA, American Society of Anesthesiologists; FEV, forced expiratory volume.

**Table 2 cancers-16-00184-t002:** Comparison of the postoperative drainage and hospital times among the patients from July 2021, when we started early chest tube removal on the day of surgery, to September 2023 between the two groups.

Variable	uVATS Group (*n* = 26)	RATS Group (*n* = 30)	*p*-Value
Postoperative drainage time, days, median (IQR)	1 (0–1)	0 (0–1)	0.27
Postoperative hospitalization time, days, median (IQR)	2 (2–2)	2 (1–3)	0.45

IQR, interquartile range.

**Table 3 cancers-16-00184-t003:** Details of the performed segmentectomies between uVATS and RATS groups.

Variable	uVATS Group (*n* = 77)	RATS Group (*n* = 30)	*p*-Value
Simple/complex, *n* (%)	35 (45.5)/42 (54.5)	10 (33.3)/20 (66.7)	0.28
LUL			
S1–3	12	3	
S1 + 2	3	8	
S1 + 2c	0	1	
S3	2	3	
S4–5	4	0	
LLL			
S6	6	2	
S8	3	0	
S9	0	1	
S9–10	2	0	
S10	0	1	
RUL			
S1	7	2	
S1 + 3	2	0	
S2	9	3	
S2 + S6a	1	0	
S3	5	1	
RLL			
S6	7	3	
S7–8	1	0	
S7–10	6	2	
S8	1	0	
S9–10	6	0	

LUL, left upper lobe; LLL, left lower lobe; RUL, right upper lobe; RML, right middle lobe; RLL, right lower lobe.

## Data Availability

The data underlying this article will be shared upon reasonable request to the corresponding author.

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
