# Peer review of "Comparison of the Learning Curve between Uniportal and Robotic Thoracoscopic Approaches in Pulmonary Segmentectomy during the Implementation Period Using Cumulative Sum Analysis"

_cancers, 2023, doi:10.3390/cancers16010184_

Round 1

Reviewer 1 Report

Comments and Suggestions for Authors

Thank you for submitting this interesting and informative manuscript to Cancers. I was pleased to receive it as a reviewer.

While your manuscript provides valuable insights into an important thoracic surgery topic, there are some areas that could be refined to further enhance the quality and impact of the work. Here are some respectful suggestions that could potentially improve the paper if you choose to implement them:

Introduction

- Including citations from prior studies that directly compare learning curves between different surgical techniques would bolster the understanding of the existing knowledge gap. Referencing these comparative studies could strengthen the argument regarding the need for further exploration and highlight the significance of your research in addressing this specific gap in the literature.

- Expanding the initial background on the promise of minimally invasive techniques would provide helpful context on why implementation studies are valuable for facilitating wider adoption.

Methods

- Adding more specifics on the CUSUM analysis methodology would enhance reproducibility. It would be beneficial to elaborate on the rationale behind selecting operative time as the primary focus over other surgical outcomes.

- Including inclusion and exclusion criteria details would inform readers on the patient population characteristics and enhance transparency.

Results

- Expanding Table 1 to include major comorbidities would provide a comprehensive overview of baseline clinical characteristics, potentially highlighting differences between the groups. This addition could further enrich the understanding of the studied cohorts' clinical profiles.

- Visualizing the CUSUM learning curves also in one figure could better showcase the differential trajectories over cases.

Discussion

- Expanding on the efficiency of the extended learning curve associated with VATS and discussing the cost implications of RATS could enhance the practicality and relevance of your discussion for clinicians and healthcare policymakers.

- Exploring the potential impact of the COVID-19 pandemic on your study's findings could be valuable, especially considering the adoption of these surgical techniques during a period marked by resource constraints and significant disruptions to conventional practices.

Overall, these suggestions aim to enhance the manuscript's quality and impact for clinicians and researchers considering adoption of robotic approaches for complex thoracic surgery. I believe that implementing some of the above suggestions would make your important work even stronger.

Author Response

I really appreciate your thoughtful comments for our manuscript.

The replies for your comments were below:

Comment 1) Including citations from prior studies that directly compare learning curves between different surgical techniques would bolster the understanding of the existing knowledge gap. Referencing these comparative studies could strengthen the argument regarding the need for further exploration and highlight the significance of your research in addressing this specific gap in the literature.

Answer1) I appreciate your valuable suggestion. However, to our best knowledge, there have been no studies comparing the learning curve between two different approaches in minimally invasive thoracic surgery.

Change in the text 1) It would be the current form.

Comment 2) Expanding the initial background on the promise of minimally invasive techniques would provide helpful context on why implementation studies are valuable for facilitating wider adoption.

Answer 2) I appreciate your valuable suggestion. A new sentence about it is added.

Change in the text 2) Please see lines 12-14 on page 6.

Methods

Comment 3) Adding more specifics on the CUSUM analysis methodology would enhance reproducibility. It would be beneficial to elaborate on the rationale behind selecting operative time as the primary focus over other surgical outcomes.

Answer 3) Your suggestion is very reasonable. Operative time was frequently used to evaluate the learning curve of several types of surgeries, which was mentioned in the discussion section.

Change in the text 3) It would be the current form.

Comment 4) Including inclusion and exclusion criteria details would inform readers on the patient population characteristics and enhance transparency.

Answer 4) Thank you for your suggestion. In this study, the consecutive cases HI operated in each approach were included without specific exclusion, which was described in the patients and methods section.

Change in the text 4) It would be the current form. 

Results

Comment 5) Expanding Table 1 to include major comorbidities would provide a comprehensive overview of baseline clinical characteristics, potentially highlighting differences between the groups. This addition could further enrich the understanding of the studied cohorts' clinical profiles.

Answer 5) Thank you for your suggestion. Only two cases in the uVATS group had postoperative morbidity, consisting of paroxysmal atrial fibrillation and prolonged air leak.

Change in the text 5) Please see lines 13-14 on page 13.

Comment 6) Visualizing the CUSUM learning curves also in one figure could better showcase the differential trajectories over cases.

Answer 6) Although I understand your suggestion, I also would like to show the raw data in each group.

Change in the text 6) It would be the current form.

Discussion

Comment 7) Expanding on the efficiency of the extended learning curve associated with VATS and discussing the cost implications of RATS could enhance the practicality and relevance of your discussion for clinicians and healthcare policymakers.

Answer 7) Thank you for your suggestion. Unfortunately, we did not compare the cost between the two approaches. Therefore, such a comment was not included in the discussion section.

Change in the text 7) It would be the current form.

Comment 8) Exploring the potential impact of the COVID-19 pandemic on your study's findings could be valuable, especially considering the adoption of these surgical techniques during a period marked by resource constraints and significant disruptions to conventional practices.

Answer 8) Thank you for your suggestion. I think the COVID-19 pandemic did not affect the choice of the surgical approaches.

Change in the text 8) It would be the current form.

Reviewer 2 Report

Comments and Suggestions for Authors

Thank you for the opportunity to analyze your interesting article.             

In this article, authors have analyzed and compared the learning curve and perioperative outcomes between the uVATS and RATS for lung segmentectomies during the implementation period.

This topic is very interesting because, the organizational constraints of our hospitals are increasing, and as surgeons, we are being asked to "do more, with less". 

To sum: 

- the number of patients eligible for segmentectomy is increasing, and will continue to do so;

- these "non-sick" patients want less invasive treatment;

- The organization of the hospital and its OR means that the number of staff is decreasing or limited.

- Learning a new approach leads to longer operating times, so fewer patients are operated on; possibly more post-operative complications at the outset too, and therefore longer hospital length of stays.

- So, an approach that can be mastered more quickly, and that brings benefits to the patient, is of interest as part of a comprehensive management approach. 

            Concerning the introduction:

            The introduction is well written, highlighting the subject. Maybe the reason of transition from VATS to uVATS can be detailed. 

No major concerns

            Concerning the methodology:

            Population:

Fig 1 in the flow chart, why did you only include 77 uVATS patients from 103, and 30 RATS from 31? 

            Surgery:  

            Some comments:

-       Did you planned the segmentcetomy using a 3D reconstruction of the patient’s lung anatomy to analyze vascular and bronchus anatomy and to analyze oncological margins?

-       You wrote that you used to sampled lymph node but you are dealing about frozen section? 

-       Did you analyze margins on frozen section? 

-       How may patients had a pre operative diagnosis? 

-       Can you precise if it’s a lymph node dissection as recommended of a lymph node sampling ? And if it’s an oriented lymph node dissection or systematic? 

-       For the intersegmental plane delineation are you also using ICG in uVATS? With the same or an other scope? 

-       Are you using sealant at the end of the procedure? 

-       A video for each approach will be welcome. 

-       How do you assess air leak? With digital device?

-       What is the suction applied for the pleural drainage? 

Criteria:

90-day mortality is more interesting than 30-day.

Concerning the CUSUM method and the statistical analysis conducted: 

            No major concerns about it. 

            Concerning the results

            Results are well reported and clearly presented in the Tables. 

            I’ll repeat some comments made earlier:

-       How many patients had a pre operative diagnosis? 

-       For lung cancer patients, can you bring the TNM classification? The nodal up-stagging status? 

-       Can you report the number of lymph node area harvested? 

Dealing with post operative outcomes:

-       Because you are removing the chest tube early, can you report this result in hour more than in days? 

Dealing with the learning cuve:

-       Figures 3 and 4 are informative, can you make an other one with the 2 CUSUMOT of uVATS and RATS?

            Concerning the discussion:

It’s a well written discussion well documented with good references. 

Dealing with the shorter hospital length of stay, and shorter duration of chest tube drainage, it’s mainly the consequence of your chest tube management protocol used since July 2021, more than the approach? 

You have evaluated the learning curve by the operative time. That’s a good way to assess this. As you have mentioned it can be assessed by the post operative complication rate also but you couldn’t perform a CUSUM analysis of po complication with your data due to a very low complication rate. 

But could you assess the learning curve with oncological quality criteria like the number of LN area harvested, or the nodal up-stagging rate for primary lung cancer? And could you do this with the 2 most important oncological quality criteria as disease-free survival and overall survival? Is it scheduled? 

Maybe more details about the lymph node dissection can be reported according your discussion about RATS advantages. Could you do this? 

If I understood the main surgeon switch for segmentectomy his approach from mVATS to uVATS and to RATS. Why? Oncological reason? Ergonomy? Unit organization? 

Limitations are well described. As you mention, previous experience could induce bias. But that’s real life. Each case enhance experience whatever the approach! 

Concerning the conclusion:

            It’s a well written, easy reading and interesting article, that need some precisions.

Maybe in your time, the same analysis can be done with one or two other surgeons? Or for all the time? Do you have any comments about this? This will brong some data about junior surgeons education and training. 

Author Response

I really appreciate your thoughtful comments for our manuscript.

The replies for your comments were below:

Concerning the introduction:

Comment 1) The introduction is well written, highlighting the subject. Maybe the reason of transition from VATS to uVATS can be detailed. 

Answer 1) Your comment is reasonable. The sentences in lines 7-13 on page 5 showed the reason why mVATS was gradually transitioned to uVATS.

Change in the text 1) It would be the current form.

Concerning the methodology:

Comment 2) Population: Fig 1 in the flow chart, why did you only include 77 uVATS patients from 103, and 30 RATS from 31? 

Answer 2) Thank you for your question. Other cases were operated by other surgeons during the study period. In this study, the two approaches were compared among the cases operated by a single surgeon (HI).

Change in the text 1) It would be the current form.

Surgery:  

Some comments:

Comment 3) Did you planned the segmentcetomy using a 3D reconstruction of the patient’s lung anatomy to analyze vascular and bronchus anatomy and to analyze oncological margins?

Answer 3) Yes, we did.

Change in the text 3) It would be the current form.

Comment 4) You wrote that you used to sampled lymph node but you are dealing about frozen section? 

Answer 4) I appreciate your suggestion. No intraoperative frozen section of the lymph nodes was examined because cT1aN0 was the criteria for intentional segmentectomies in our department.

Change in the text 4) Please see lines 9-11 on page 10.

Comment 5)  Did you analyze margins on frozen section? 

Answer 5) Yes, we did. However, the exact data was lacked for some cases in the uVATS group. Therefore, we did not compare it between the two groups.

Change in the text 5) It would be the current form.

Comment 6) How many patients had a pre operative diagnosis? 

Answer 6) Unfortunately, the data about the preoperative diagnosis were lacked.

Change in the text 6) It would be the current form.

Comment 7)  Can you precise if it’s a lymph node dissection as recommended of a lymph node sampling ? And if it’s an oriented lymph node dissection or systematic? 

Answer 7) Lymph node sampling was recommended for pathological stage confirmation.

Change in the text 7) It would be the current form.

Comment 8) For the intersegmental plane delineation are you also using ICG in uVATS? With the same or an other scope? 

Answer 8) We usually used ICG in the uVATS group except for the cases with iodine allergy. And, the same scope was adopted because the thoracoscopy had an additionla infrared mode.

Change in the text 8) It would be the current form.

Comment 9) Are you using sealant at the end of the procedure? 

Answer 9) If an air leak was detected during a seal test at the end of surgery, the leak was sutured with an absorbable monofilament or sealant was applied.

Change in the text 9) Please see lines 3-5 on page 10.

Comment 10) A video for each approach will be welcome. 

Answer 10) Thank you for your suggestion. However, due to the limitation of the word count, the video was not involved.

Change in the text 10) It would be the current form.

Comment 11) How do you assess air leak? With digital device?

Answer 11) Thank you for your suggestion. However, the digital drainage system or conventional drainage bottle was applied.

Change in the text 11) It would be the current form.

Comment 12)  What is the suction applied for the pleural drainage? 

Answer 12) Suction was applied for any case.

Change in the text 12) It would be the current form.

Comment 13) 90-day mortality is more interesting than 30-day.

Answer 13) Thank you for your suggestion. 90-day mortality was added.

Change in the text 13) Please see the revised table 1.

Comment 14)  For lung cancer patients, can you bring the TNM classification? The nodal up-stagging status? 

Answer 14) Although I can show the data, it might not suitable in this study because the number of patients with primary lung cancer was only 19 in the RATS group.

Change in the text 14) It would be the current form.

Comment 15)  Can you report the number of lymph node area harvested? 

Answer 15) Although I can show the number of harvested lymph nodes, it might not suitable in this study because we performed not radical lymphadenectomy but sampling.

Change in the text 15) It would be the current form.

Comment 16)  Because you are removing the chest tube early, can you report this result in hour more than in days? 

Answer 16) Thank you for your question. Indications for drain removal on the day of surgery were absence of air leakage during an intraoperative seal test, radiographic evidence of lung expansion, and continuous absence of air leakage through a drainage bottle for 2-4 hours after surgery.

Change in the text 16) Please see lines 1-4 on page 11.

Comment 17) Figures 3 and 4 are informative, can you make an other one with the 2 CUSUMOT of uVATS and RATS?

Answer 17) Thank you for your suggestion. Please compare the learning curve of CUSUMOT using figure 3 and 4.

Change in the text 17) It would be the current form.

Comment 18) Dealing with the shorter hospital length of stay, and shorter duration of chest tube drainage, it’s mainly the consequence of your chest tube management protocol used since July 2021, more than the approach? 

Answer 18) I think so because the postoperative drainage and hospitalization time were similar between the two groups after the introduction of early removal of postoperative drain (table 2).

Change in the text 18) It would be the current form.

Comment 19) But could you assess the learning curve with oncological quality criteria like the number of LN area harvested, or the nodal up-stagging rate for primary lung cancer? And could you do this with the 2 most important oncological quality criteria as disease-free survival and overall survival? Is it scheduled? 

Answer 19) I totally agree with the opinion that we can harvested more lymph node via RATS than via uVATS. However, in pulmonary segmentectomy, we performed not radical lymphadenectomy but sampling, which was not suitable to investigate which approach was superior in the lymphadenectomy. As you said, I will conduct the study to investigate the long-term results in the future after gaining the cases.

Change in the text 19) It would be the current form.

Comment 20) Maybe more details about the lymph node dissection can be reported according your discussion about RATS advantages. Could you do this? 

Answer 20) I totally agree with the opinion that we can harvested more lymph node via RATS than via uVATS. However, in pulmonary segmentectomy, we performed not radical lymphadenectomy but sampling, which was not suitable to investigate which approach was superior in the lymphadenectomy.

Change in the text 20) It would be the current form.

Comment 21) If I understood the main surgeon switch for segmentectomy his approach from mVATS to uVATS and to RATS. Why? Oncological reason? Ergonomy? Unit organization? 

Answer 21) First, we changed the surgical approach from mVATS to uVATS in order to explore less invasiveness. Second, I started RATS due to the unit organization.

Change in the text 21) It would be the current form.

Comment 22) Maybe in your time, the same analysis can be done with one or two other surgeons? Or for all the time? Do you have any comments about this? This will brong some data about junior surgeons education and training. 

Answer 22) Thank you for your comment. Unfortunately, the number of cases other surgeons operated via RATS was too few. Therefore, it is difficult to perform the comparison so far.

Change in the text 22) It would be the current form.